# Uba1: A Potential Ubiquitin-like Activator Protein of Urm1 in *Toxoplasma gondii*

**DOI:** 10.3390/ijms231810298

**Published:** 2022-09-07

**Authors:** Qianqian Xiao, Jinxuan Li, Junpeng Chen, Qianqian Tan, Xiao Chen, Hongmei Li, Xiaomin Zhao, Xiao Zhang

**Affiliations:** 1Room 115, Department of Preventive Veterinary Medicine, College of Veterinary Medicine, Shandong Agricultural University, Tai’an 271018, China; 2Shandong Provincial Key Laboratory of Animal Biotechnology and Disease Control and Prevention, Shandong Agricultural University, Tai’an 271018, China; 3Shandong Provincial Engineering Technology Research Center of Animal Disease Control and Prevention, Shandong Agricultural University, Tai’an 271018, China

**Keywords:** *Toxoplasma gondii*, Urm1, ubiquitin-like activator

## Abstract

We had shown in our previous study that TgUrm1 (ubiquitin-related Modifier 1) was involved in the regulation of anti-oxidant stress in *Toxoplasma gondii* by conjugating with TgAhp1. It is generally believed that Urm1 binds to target proteins through a mechanism involving Uba (ubiquitin-like activator protein). Here, we identified the TgUrm1-exclusive ubiquitin-like activator-TgUba1, which was located in the cytoplasm of *Toxoplasma*. TgUba1 contained three domains, including the atrophin-1 domain (ANT1), the E1-like domain (AD), and the rhodanese homology domain (RHD). We explored the interaction of TgUba1 with TgUrm1, and the AD domain was essential for the interaction of the two proteins. The TgUba1 knockout and complementary mutants were obtained based on CRISPR/Cas9 gene editing technology. The knockout of TgUba1 attenuated parasite proliferation and virulence in mice, but not invasion and egress processes, revealing the pivotal role played by TgUba1 in *T. gondii* survival. Meanwhile, the conjugate band of TgUrm1 was significantly reduced under oxidative stress stimulation without TgUba1, indicating that TgUba1 enhanced the targeted conjugation ability of TgUrm1 in response to oxidative stress, especially under diamide (Dia) stimulation. Furthermore, eleven TgUba1-interacting proteins were identified by proximity-based protein labeling techniques, relating them to ubiquitin-like modifications, anti-oxidative stress and metabolic regulation processes. In conclusion, TgUba1 was essential for *T. gondii* survival and might be a potential ubiquitin-like activator protein for TgUrm1.

## 1. Introduction

*Toxoplasma gondii* is an obligate intra-cellular eukaryotic parasite of the phylum Apicomplexa [1]. As a zoonotic pathogen, it can flow with the blood and destroy vital organs, such as the heart and brain [2]. Generally, most infections are asymptomatic, and sometimes *T. gondii* can cause retinal lesions and life-threatening central nervous system infections in immunocompromised individuals and the developing fetus [3].

Urm1, a metabolic regulatory system with dual functions as a protein modifier and sulfur carrier [4], had been found to be involved in metabolic regulation by covalently conjugating with lysine residues of other proteins in recent years [5,6]. So far, relatively few studies had been conducted on Urm1, which was more common in *Saccharomyces cerevisiae* and had not been reported in parasitic protozoa [7]. Our laboratory previously identified Urm1 in *T. gondii* and found that deletion of Urm1 resulted in reduced tachyzoite invasion, proliferation and virulence in mice, confirming its essential importance for the survival of *T. gondii*. Oxidative stress was found to be a strong inducer of TgUrm1 conjugation [8]. Under oxidative stress, the interaction of TgUrm1 with the peroxiredoxin TgAhp1 exerted an anti-oxidative effect in *T. gondii* [9]. In addition, TgUrm1 seemd to be able to guide *T. gondii* to escape immune clearance by the host [8].

Urm1 should be activated before binding to substrate proteins. However, studies on the TgUrm1 ubiquitination pathway have not been thoroughly explained. In general, UBLs (ubiquitin-like molecules) follow a classical activation pathway, catalyzed by an ATP-dependent E1-E2-E3 conjugation cascade, which leads to the implementation of various basic functions within the cell [10,11]. Other UBLs follow a non-canonical activation pattern involving only one or more of the E1, E2 and/or E3 enzymes in the ubiquitin-like process [12,13]. It is generally accepted that Urm1 of *S. cerevisiae* follows a non-canonical activation pathway, which requires modification and activation (thiocarboxylation cycle) of ScUba4 before it can exert its molecular function [14,15]. ScUba4 contained two domains for ubiquitin-like modification: a canonical E1-like domain (AD), which activates Urm1 by forming a ubiquitin-thioester intermediate, and a rhodanese homology domain (RHD) [14,16]. This sulfurtransferase domain catalyzed the formation of a C-terminal thiocarboxylate on Urm1 [14,17]. On the basis of our previous report [8], a special gene (TGGT1_212100) with AD and RHD conserved domains was retrieved, which might be a potential activator of TgUrm1.

In this study, we identified the Uba1 gene in *Toxoplasma* and explored the key domain of TgUba1 that interacted with TgUrm1, indicating the exceptional possibility of TgUba1 as a potential ubiquitin-like activator of TgUrm1.

## 2. Results

### 2.1. Characterization of Uba1 in T. gondii

Based on our previous report, a special gene (TGGT1_212100) was screened, which encoded a protein consisting of 1076 amino acid residues. We obtained the full-length coding sequence by PCR using RH cDNA (Figure 1A). Phylogenetic analysis showed high homology with *Saccharomyces cerevisiae*, *Kluyveromyces marxianus* and *Candida albicans* (Figure 1B). The protein sequence of this putative TgUba1 was analyzed on the SMART MODE and National Center for Biotechnology Information (NCBI) online websites, revealing three conserved domains, including the atrophin-1 domain (ANT1), the E1-like domain (AD) and the rhodanese homology domain (RHD) (Figure 1C). The AD domain contained two conserved CxxC-motifs, which were regions with conserved cysteines corresponding to the canonical site of thioester formation in UBL systems. Within the C-terminal RHD domain, the CRRGND active loop motif was also included. Multiple sequence alignment (MSA) analysis of Uba confirmed the fact that all features of a typical Uba was displayed by TgUba1, including two conserved cysteine sites within the AD and RHD domains (Figure 1D).

To assess the expression and localization of TgUba1 in tachyzoites, a truncated TgUba1 protein fused to a histidine tag was successfully expressed in *E. coli* and used to generate anti-TgUba1 serum in mice. The anti-TgUba1 polyclonal antibody reacted with the 135 kDa protein in the tachyzoite lysate by western blotting (Figure 1E). TgUba1 was specifically recognized by the polyclonal antibody and revealed a cytoplasmic localization of TgUba1 (Figure 1F), which was consistent with Uba in other organisms. Therefore, TGGT1_212100 was designated as TgUba1.

### 2.2. Confirmation of the Interaction of TgUba1-TgUrm1

To study the functional connection between TgUba1 and TgUrm1, we first confirmed the interaction between TgUba1 and TgUrm1 using yeast two-hybrid experiments. As shown in Figure 2A, the prey TgUrm1 and bait TgUba1 strains appeared blue on DDO/X/A plates. To exclude false positives, each positive clone isolated by colony PCR was transferred to the QDO/X/A plate for further cultivation and the colonies remained blue, indicating a positive two-hybrid interaction between TgUba1 and TgUrm1. Furthermore, Co-IP assay performed in 293T cells demonstrated that TgUba1 successfully non-covalently immunoprecipitated TgUrm1 protein, indicating a specific interaction between TgUba1 and TgUrm1 (Figure 2B).

### 2.3. AD Domain as the Key Domain Targeting TgUrm1

To further investigate the key domain mediating the interaction of TgUba1 between TgUrm1, four truncated versions of TgUba1 were generated, encompassing various regions of the key domains. The AD domain and the complex AD and RHD domains remained blue on the QDO/X/A plate, preserving the two-hybrid interaction. In contrast, colonies transformed with the other two domains showed negative interactions (Figure 3A). Co-IP assay also supported this result. The TgUrm1 protein was successfully non-covalently captured by the AD domain in transfected 293T cells, indicating an interaction relationship between the AD domain and TgUrm1, while the RHD domain completely abolished the interaction (Figure 3B). Thus, our observations identified a crucial role of the AD region in interaction with TgUrm1.

### 2.4. Construction of ΔUba1 and CM-Uba1 Strains by CRISPR/Cas9

To test the function of TgUba1 in the parasite, TgUba1 was disrupted in the RHΔKu80 strain, resulting in the knockout strain RHΔUba1 and the complementary strain CM-Uba1 (Figure 4A). On the basis of the complete knockout failure, we used the CRISPR/Cas9 strategy to generate an insertional knockout strain by replacing the 0.1 kb genomic region encompassing the TgUba1 open reading frame with DHFR. This demonstrated that TgUba1 is not an essential gene for *T. gondii* and that the failure of complete loss of TgUba1 might be caused by the structural irreplaceability of the gene on the chromosome. Meanwhile, the original genes’ existence was not detected by diagnostic PCR (Figure 4B) and the proteins’ expression was not observed by western blot (Figure 4C) and IFA (Figure 4D), confirming that we have obtained the knockout and complementary expression of tachyzoites.

### 2.5. Parasites Lacking TgUba1 Showed Significantly Reduced Proliferation and Virulence in Mice

Obvious plaque differences were observed in parasites of ΔUba1 strain and the CM-Uba1 strain (Figure 5A). The results of the plaque area showed a statistically significant difference in the mean values of parasite formation between wild-type ΔKu80 and ΔUba1, but no significant difference in the total quantity. Briefly, plaque experiments demonstrated that ΔUba1 was defective in parasite proliferation, but not in invasion. Invasion and proliferation assays (Figure 5B,C) confirmed that TgUba1 only affected the proliferation process of *T. gondii*, which was consistent with the plaque assay results. Additionally, the egress rate (Figure 5D) and gliding rate (Figure 5E) were quantified for all strains. Our results illustrated that ΔUba1 parasites glide flawlessly compared to ΔKu80 and that there was no significant change in parasite egress. Subsequently, the pathogenicity of the ΔUba1 strain was evaluated in mice. Mice infected with ΔKu80 and CM-Uba1 parasites developed clinical symptoms, including abdominal swelling and fur ruffling, on day 5 post-infection and died between days 7 and 8. In contrast, mice challenged with ΔUba1 parasites all died between days 9 and 10. Significant differences between ΔUba1 and ΔKu80/CM-Uba1 parasites (Figure 5F) were observed. In general, disruption of TgUba1 reduced *T. gondii* proliferation and pathogenicity in mice without affecting invasion and gliding motility.

### 2.6. TgUba1 Enhanced the Target Conjugation Ability of TgUrm1 upon Oxidative Stress

We constructed an overexpressed Urm1 strain based on ΔUba1 by using CRISPR/Cas9 technology (Figure 6A). Monoclonal strains were identified by PCR (Figure 6B), western blotting (Figure 6C) and IFA (Figure 6D). Treating tachyzoites with different oxidative stressors, a unique pattern of high molecular weight polypeptides was observed in TgUba1-present strains, especially when stimulated with 600 μM of diamide (Dia), suggesting that TgUba1 enhanced the targeting ability of TgUrm1 to proteins in *Toxoplasma* under conditions of oxidative stress (Figure 6E).

### 2.7. Screening for Other Interacting Proteins of TgUba1 by TurboID

Except for the two classical domains, TgUba1 had a special domain-ANT1. To investigate other functions exerted by TgUba1 in *T. gondii*, a CM-Uba1-TurboID strain expressing a TurboID fusion protein was generated (Figure 7A). PCR showed a correct integration of TurboID at the C-terminus of TgUba1 (Figure 7B). A 173kDa band was observed in the western blot, indicating the fusion expression of TgUba1 and TurboID (Figure 7C). In Figure 7D, the interacting proteins of TurboID-tagged TgUba1 were detected to accumulate in the cytoplasm. Meanwhile, silver staining tests and western blotting analysis showed that most of the biotinylated proteins were captured from the whole lysate (Figure 7E,F). After the removal of common contaminants and comprehensive analysis, a total of 24 proteins were identified by mass spectrometry, and eleven promising proteins were selected for ubiquitination modification, metabolic regulation and anti-oxidative stress, including TgUrm1, which has been validated for interactions (Figure 7G).

## 3. Discussion

TgUba1 was first identified in *T. gondii* and appeared to be cytoplasmically localized. The absence of TgUba1 resulted in a marked reduction in the intracellular proliferative capacity of the parasite and virulence in mice. TgUba1 interacted with TgUrm1, and the AD domain played a key role in this process. Meanwhile, TgUba1 enhanced the target conjugation ability of TgUrm1 upon oxidative stress, suggesting that TgUba1 might be a potential ubiquitin-like activator of TgUrm1 with a special possibility and was most likely involved in the ubiquitin-like modification, metabolic regulation and anti-oxidative stress processes of *T. gondii*.

Our laboratory had previously identified TgUrm1 and demonstrated its important role in the regulation of oxidative stress in *T. gondii*. Ubiquitination in *S. cerevisiae* had been shown to be dependent on the Urm1-specific E1 enzyme Uba4 [18], also known as molybdenum cofactor synthesis 3 (MOCS3) in mammals [17]. In this study, we showed TgUba1 (TGGT1_212100), a potential Ubiquitin-like (UBL) activator protein of TgUrm1. Bioinformatics analysis revealed that TgUba1 belonged to the superfamily of ubiquitin-like protein activating enzymes (E1) and shared up to 38% of protein homology with ScUba4. Three typical domains were revealed in this protein, among which the AD and RHD domains used for ubiquitin-like modifications were also possessed by ScUba4. CxxC-motifs chelate zinc ions, which were regions containing conserved cysteines corresponding to the canonical site for thioester formation in UBL systems [17]. The C-terminal RHD domain had been reported to thiocarboxylate Urm1 [14], while the function of the N-terminal ANT1 domain remained un-known. These pieces of evidence suggested that TgUba1 had the structural basis for modifying TgUrm1.

In some eukaryotes, such as *S. cerevisiae*, the ubiquitin-like activator of Urm1, Uba, covalently activated Urm1 through the synergy of the AD and RHD domains [14]. Studies had revealed that the carboxyl group of the C-terminal glycine of ScUrm1 formed a covalent thioester bond with a typical cysteine, which was located in the cross-loop of the ScUba4 AD domain [19]. ScUrm1 then formed an acyl-persulfide bond with cysteines in the RHD. In vivo re-generation of ScUba4 was accompanied by the release of thiocarboxylated ScUrm1 [14]. The thioester formation was a central but not absolute step in the classical UBL cascade [19,20]. In this study, we had demonstrated that only the AD domain played a key role in the interaction between TgUba1 and TgUrm, which seemed to be contrary to previous reports. Whether the RHD domain, which did not interact with TgUrm1, acted as the E2 enzyme of TgUrm1 was un-known. We speculated that the activation process of TgUba1-TgUrm1 might follow an un-known mode, at least different from other eukaryotes, and possibly involved the synergy of other activating enzymes. Taken together, these results provided ample evidence that TgUba1 had a molecular function in activating and modifying TgUrm1, but the specific activation process was not yet understood.

Strikingly, ScUrm1 and ScUba4 were covalently linked [14]. However, Co-IP revealed a solitary TgUrm1 molecule rather than a TgUba1-TgUrm1 conjugation, indicating the formation of a non-covalent TgUba1-TgUrm1 conjugation. TgATG12 was the same as TgUrm1, which was also a ubiquitin-like molecule recently identified in *T. gondii*. Un-like other species, the targeting mode of TgATG12 with interacting proteins had evolved from covalent to non-covalent, retaining its original function in *T. gondii* [21]. Meanwhile, previous studies had shown that urmylation of TgUrm1 did not depend exclusively on the sulfide modification of TgUba1 [8]. Thus, the key role of TgUba1 in TgUrm1 urmylation remained to be explored.

It is generally believed that the process of activation of Urm1 by Uba is a short-term process, and is followed by the separation of the two proteins, indicating that TgUba1 may have other functions besides activating TgUrm1. The adjacent biotin-labeling technique was used to screen for proteins that interacted with TgUba1 and identified a series of proteins. TGGT1_209000 appeared to be an E3 enzyme in the ubiquitin-like process. TGGT1_219820 was a ubiquitin-like molecule similar to TgUrm1, and the FkpA domain of TGGT1_283850 was analyzed to be involved in post-translational modification. Furthermore, several anti-oxidative stress proteins were identified, including the thiol-dependent peroxidase Prx3 (TGGT1_230410) [22], but the possibility that they were TgUrm1 targets cannot be ruled out.

The ubiquitin-like system had been shown to be involved in mediating protein lipidation [23], either as E2 enzymes [24] or as E3 enzymes [25]. Notably, the involvement of ScUba4 in the lactic acid adaptation mechanisms and metabolic processes of nitrogen decomposition via urmylation with Urm1 provided a theoretical basis for the study of the involvement of TgUba1 in the regulation of metabolic processes [26,27]. Combined with the identification of metabolism-related proteins by mass spectrometry results, our study revealed a low proliferation rate in cells and virulence in mice of the ΔUba1 strain, speculating that TgUba1 might be involved in some metabolic pathways that determined the growth of *T. gondii*. As shown in Figure 7G, the 14-3-3 protein (TGGT1_263090) and SPX domain-containing protein (TGGT1_248550) were associated with phosphate metabolism [28,29]. Although the functional link between the ubiquitin-like processes involved in TgUba1 and metabolism had not been established, this did not interfere with the speculation that TgUba1 might also be involved in metabolic regulation.

## 4. Materials and Methods

### 4.1. Cells and Parasites

The *T. gondii* Type I RH ΔKu80 strain was maintained in African green monkey kidney (Vero) cells in Dulbecco’s minimum essential medium (DMEM) (M&C gene Technology Ltd., Beijing, China) supplemented with 2% fetal bovine serum at 37 °C and 5% CO_2_.

### 4.2. Intracellular Invasion and Replication Assay

Parasite growth rate was assessed by counting the number of parasites per vacuole. Purified parasites were inoculated into monolayers of DF-1 cells in 12-well plates (5 × 10^4^ parasites per well) and were allowed to invade cells under normal growth conditions for 2 h. DF-1 cell monolayers were then washed 3 times with PBS to remove extracellular parasites. At the designated time after infection, tachyzoites in vacuoles were marked with rabbit anti-GAP45 by IFA and counted in at least 100 vacuoles. Experiments were conducted in triplicate and were repeated three times.

### 4.3. Egress Assay

DF-1 cells were infected with 5 × 10^4^ tachyzoites and cultured under normal growth conditions for 24 h. Monolayers were washed three times in PBS to remove extracellular parasites and then incubated with 10 μM A23187 (Sigma, St. Louis, MO, USA) (a calcium ionophore) for 5 min. After incubation, parasites were fixed, and IFA was performed with rabbit anti-GAP45 to measure the percentage of intact and egressed PVs. A minimum of 100 vacuoles were counted per slip.

### 4.4. Pathogenicity of T. gondii in Mice

Four-week-old female BALB/c mice, purchased from Jinan Pengyue Experimental Animal Breeding Co., Ltd. (Jinan, China), were allowed to acclimate in our facility for 1 week. Freshly purified tachyzoites were injected intraperitoneally at a dose of 500/mice. Five mice were infected in each group, and the survival and symptoms of the mice were monitored daily. The virulence test was repeated three times.

### 4.5. Co-Immunoprecipitation Analysis

Human embryonic kidney-derived 293T cells were co-transfected with bait and prey plasmids. Cells were harvested 48 h after transfection and lysed in weak RIPA buffers for 30 min. Cell lysates were centrifuged at 12,000 rpm for 10 min at 4 °C. Supernatants were harvested and incubated with GFP Affinity Magnetic Beads for 6 h at 4 °C with gentle rotation. The beads were washed with 30% weak RIPA and collected with magnetic stand. The beads were then re-suspended in an SDS-PAGE sample buffer. Samples were boiled, loaded onto gels for SDS-PAGE, and then analyzed by western blotting (WB).

### 4.6. Yeast Two-Hybrid Screen

Before starting a two-hybrid screen, it is necessary to test the self-activation of pGBKT7-TgUba1 on the DO/X (SD/-Trp) plate. By including a minimal effective dose of Aba in the yeast media plates used for screening, you will increase the likelihood of identifying weak protein–protein interactions in yeast media plates lacking Trp. The Y2HGold strain was transformed with ligated pGBKT7-TgUba1 and pGADT7-TgUrm1 from yeast genomic two-hybrid libraries. Transformants were selected to be grown on DDO/X/A (SD/-Leu/-Trp). After 5–7 days, the emerging blue colonies were picked out and inoculated onto fresh QDO/X/A (SD/-Leu/-Trp/-Ade/-His) plates. An insert of each positive clone was isolated by colony PCR and identified by DNA sequencing.

### 4.7. Enrichment and Identification of Biotinylated Proteins

We utilized a proximity labeling strategy that used the biotin ligase, TurboID, to efficiently label the proteome in ΔUba1 tachyzoites for specific complementation. We created stable Uba1-TurboID tachyzoites. TurboID-expressing tachyzoites were seeded in 15 cm Petri dishes and cultured overnight. Biotin (final concentration of 500 μm) was added to the medium. After 24 h of incubation, the tachyzoites were washed twice with PBS and lysed in strong RIPA. The lysates were centrifuged at 12,000 rpm for 15 min at 4 °C. The supernatants were recovered and enriched using streptavidin magnetic beads (Beaver Biosciences Inc., Guangzhou, China). The beads were washed 15 times with strong RIPA and then 8 times with 8 M urea. Finally, the beads were magnetically separated, and the elution buffer containing the biotinylated proteins was transferred to a new 1.5 mL EP tube. The sample was electrophoresed on 12% SDS-PAGE and the differential bands were identified by mass spectrometry.

### 4.8. Statistical Analysis

Independent experiments were conducted at least three times, with a minimum of three technical replicates for each experiment. All graphs and statistical analyses were generated using GraphPad Prism 8 (GraphPad Software, San Diego, CA, USA, www.graphpad.com). Statistical data were expressed as the mean value of the standard error of the mean (SEM). All analyses were performed with a two-tailed Student’s *t*-test, except for the parasite proliferation assay and mice virulence assay, which were analyzed with a two-way ANOVA and the Gehan-Breslow-Wilcoxon test. *p* < 0.05 was considered statistically significant.

## Figures and Tables

**Figure 1 ijms-23-10298-f001:**
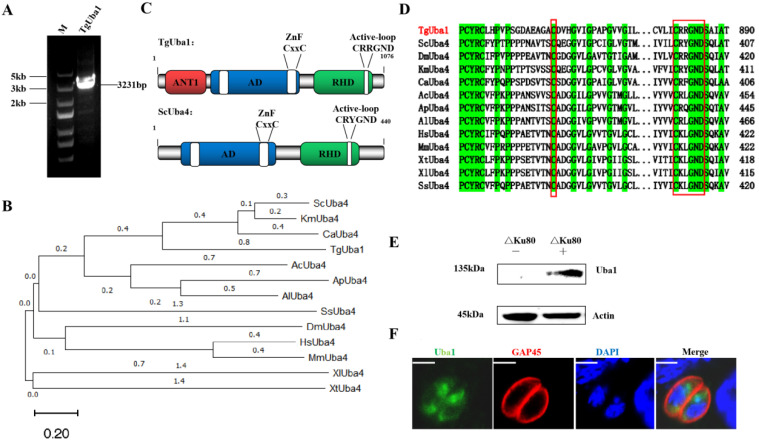
Identification and cellular localization of TgUba1. (**A**) PCR amplification of TgUba1. The full length is 3231 bp. (**B**) Phylogenetic analysis of TgUba1 with MEGA 11.0 tool. The sequence accession numbers used were TGGT1_212100 (Protista), *Saccharomyces cerevisiae* (Fungi) NC_001140.6, *Drosophila melanogaster* (Animalia) NT_033779.5, *Kluyveromyces Marxianus* (Fungi) NC_036026.1, *Candida Albicans* (Fungi) NC_032089.1, *Aspergillus chevalieri* (Fungi) NC_057365.1, *Aspergillus Puulaauensis* (Fungi) NC_054858.1, *Aspergillus luchuensis* (Fungi) NC_054852.1, *Homo Sapiens* (Animalia) NC_000020.11, *Mus Musculus* (Animalia) NC_000068.8, *Xenopus tropicalis* (Animalia) NC_030681.2, *Xenopus laevis* (Animalia) NC_054379.1 and *Sus scrofa* (Animalia) NC_010459.5. (**C**) Conserved domain analysis of TgUba1. (**D**) Multiple sequence alignment of TgUba1. Green-filled rectangular frames showed conserved amino acids; red rectangular frames indicated the conserved cysteine—the site corresponding to the formation of thioesters in the UBLs system, and the active loop motif; dots indicated gaps or missing residues. (**E**) Western blot analysis of native TgUba1. Total antigens from cell-cultured RHΔKu80 tachyzoites. The expected bands were elicited by the anti-TgUba1 polyclonal antibody. “−” represents negative serum, and “+” represents TgUba1 post-immunization serum. (**F**) IFA analysis of TgUba1 localization. TgUba1 (green) and TgGAP45 (red) in RH∆Ku80 determined the localization. Scale bar, 2.5 μm. DAPI, 4′,6-diamidino-2-phenylindole, nuclear dye. TgGAP45, gliding-associated protein 45.

**Figure 2 ijms-23-10298-f002:**
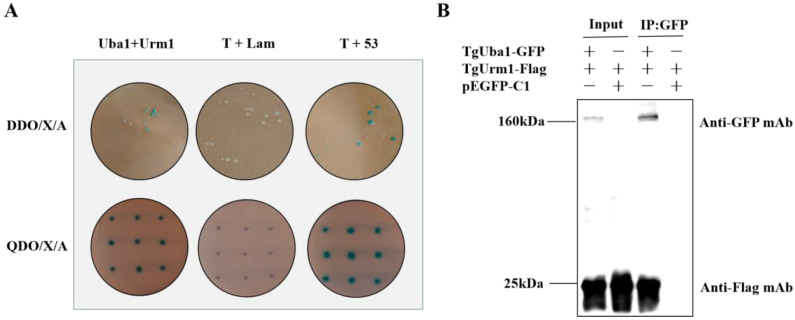
The binding of TgUba1 and TgUrm1. (**A**) Yeast two-hybrid assay using TgUba1 as a bait protein demonstrated the interaction between TgUba1 and TgUrm1. Co-transformation of pGBKT7-Lam and pGADT7-T served as a negative control, while co-transformation of pGBKT7-53 and pGADT7-T served as a positive control. DDO/X/A, medium lacking leucine and tryptophan. QDO/X/A, medium lacking leucine, tryptophan, adenine and histidine. X-α-Gal was used for chromogenic positive colonies, and AbA was used for growth screening. (**B**) Co-IP assay was conducted to verify the interaction between TgUba1 and TgUrm1. TgUba1 was immunoprecipitated from 293 T cell lysates with GFP antibody and TgUrm1 was immunoprecipitated with flag antibody.

**Figure 3 ijms-23-10298-f003:**
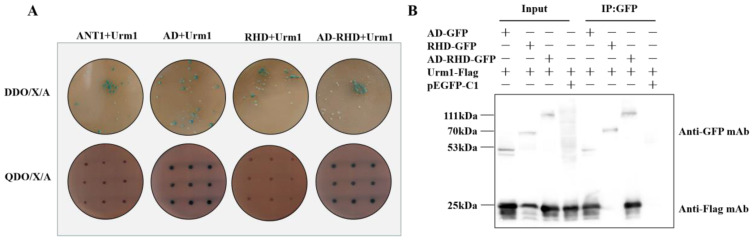
Identification of AD interacting with TgUrm1. (**A**) Yeast two-hybrid analysis of domain interactions. The AD, as well as the complex AD and RHD domains, showed positive yeast two-hybrid interactions with TgUrm1. (**B**) Co-IP analysis of 293 T cells transfected with the indicated plasmids.

**Figure 4 ijms-23-10298-f004:**
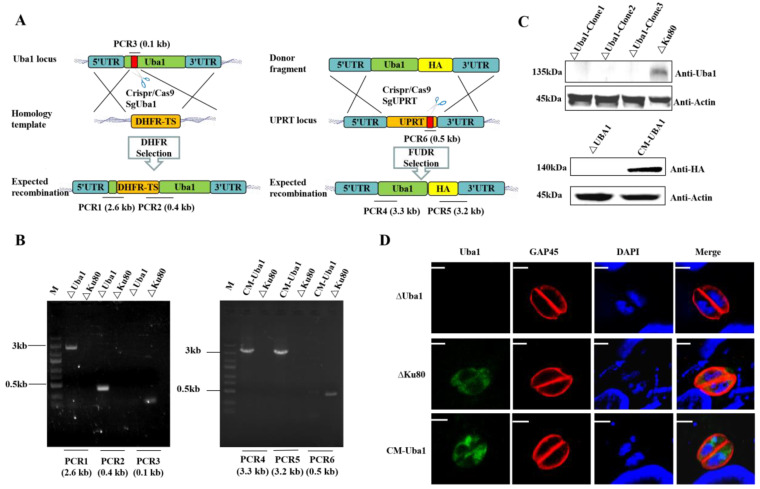
Construction of TgUba1 gene editing strains. (**A**) Schematic diagram of CRISPR/CAS9-mediated disruption of knockout and complementary strain construction. The red bar indicated the sgRNA target region. (**B**) PCR verification of TgUba1 deletion and complementation. PCR1, PCR2 and PCR4, PCR5 checked for 5′ and 3′ integration of the selection marker, while PCR3 and PCR6 examined successful deletion and complementation of TgUba1 gene. (**C**) Western blot verification of deletion and complementation of TgUba1. (**D**) IFA verification of TgUba1 loss and complementation. PVs were marked in red by rabbit anti-GAP45 polyclonal antibody, while TgUba1 was marked in green by anti-TgUba1 polyclonal antibody. Scale bar, 2.5 μm.

**Figure 5 ijms-23-10298-f005:**
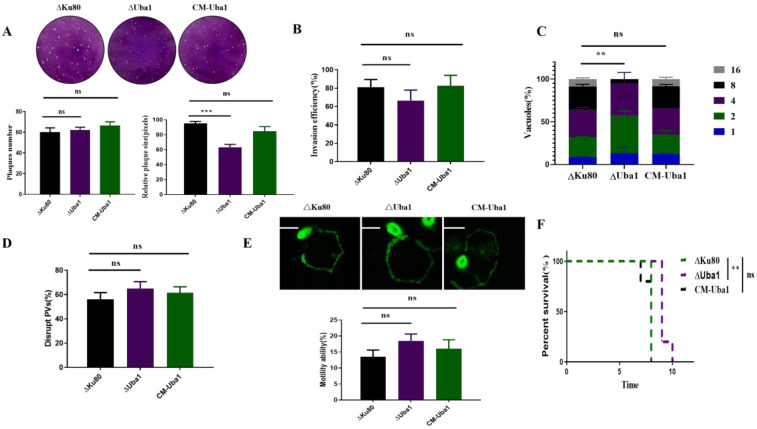
Phenotypic consequences of TgUba1 disruption. (**A**) Plaque assay of ΔUba1 parasites. Parasites were added to the DF-1 monolayer for 8 days and the size of the lysis plaques was measured. Values were the mean standard error of the mean value and area of plaques (30 plaques were measured in each condition) from one of the two representative experiments. ns, *p* > 0.05, ***, *p* < 0.001. (**B**) Invasion assay of CM-Uba1 and ΔUba1 parasites. n = 3, a representative experiment from two independent assays. ns, *p* > 0.05, indicates non-significant. (**C**) Proliferation assay of CM-Uba1 and ΔUba1 parasites. n = 3, a representative experiment from 2 independent assays. ns, *p* > 0.05. **, *p* < 0.01. (**D**) Percentage egress of each strain calculated following treatment with A23187 at 24 h post-infection. n = 3, ns, *p* > 0.05. (**E**) Movement trajectory statistical data. Gliding was observed using IFA as the surface protein SAG1, which was left behind as the parasite glides. ns, *p* > 0.05. Scale bar, 5 μm. (**F**) Survival curves of mice being infected with maternal ΔKu80, ΔUba1 or CM-Uba1 parasites at a dose of 500/mice. The results of one of three similar experiments were shown and analyzed by the Gehan–Breslow Wilcoxon test. ns, *p* > 0.05. **, *p* < 0.01.

**Figure 6 ijms-23-10298-f006:**
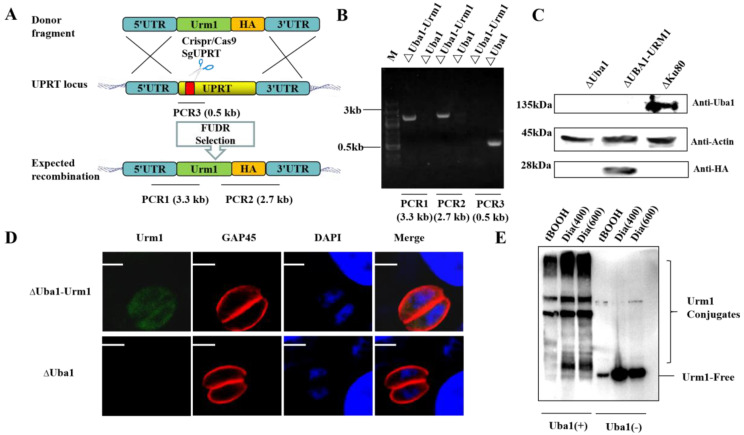
Effects of TgUba1 on TgUrm1 oxidative conjugation. (**A**) Schematic representation of TgUrm1-HA mediated UPRT gene deletion in *T. gondii*. The red bar indicated the sgRNA target region. (**B**) Performance of confirmatory PCR using genomic DNA as template from ΔUba1 and transfected parasite clones. (**C**) Immunoblotting demonstrating expression of epitope-tagged TgUrm1-HA. (**D**) Immunofluorescence (IFA) of ΔUba1-Urm1 using anti-HA antibodies. Scale bar, 2.5 μm. (**E**) Total tachyzoite lysates resolved by SDS/PAGE and subjected to anti-HA western blotting. The treatment were as follows: 10^−4^ M tBOOH for 2 h, 400 and 600 μM of Dia for 10 min. tBOOH (tert-butyl hydroperoxide) was an oxidative stress inducer. “+” indicated the presence of TgUba1 gene in parasites, and “−” indicated the absence of TgUba1 gene.

**Figure 7 ijms-23-10298-f007:**
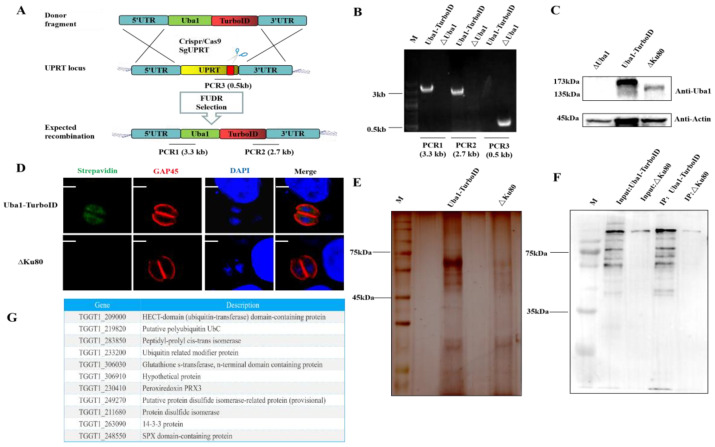
TurboID labeling of TgUba1-proximal proteins. (**A**) Tachyzoites expressing TgUba1-TurboID generated by CRISPR/Cas9-mediated genome editing. The red bar indicated the sgRNA target region. (**B**,**C**) Single-clone PCR and western-blot identification of TgUba1-TurboID. (**D**) Confocal microscopy image of tachyzoites expressing TgUba1-TurboID cultured in the presence of 500 μm biotin for 24 h. Scale bar, 2.5 μm. (**E**) Silver staining test analysis of lysates obtained from tachyzoites expressing the TgUba1-TurboID fusion protein. (**F**) Analysis of cell lysates by western blotting. (**G**) Analysis of mass spectrometry data.

## Data Availability

Data openly available in a public repository.

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
