# Peer review of "Uba1: A Potential Ubiquitin-like Activator Protein of Urm1 in Toxoplasma gondii"

_ijms, 2022, doi:10.3390/ijms231810298_

Round 1
Reviewer 1 Report
In the current study, a cytoplasmic localized protein Uba1 in Toxoplasma was identified. They demonstrate the Uba1 interact with Urm1 and is potential ubiquitin-like activator protein for TgUrm1. In addition, they also analyze the function of Uba1 in Toxoplasma replication, The work was excellent, while there are some points need to improve.
1. Line 18, this sentence is misleading, please rephrase.
2. Line 21, please delete “only”.
3. Line 23, “reduced” was used incorrect, “attenuated” is better.
4. Line 71, How can the phylogenetic analysis come to a conclusion “indicating that TgUba1 can activate and modify TgUrm1”, it’s better to delete this sentence.
5. Line 85, “multi-antibody” is misleading, please rephrase.
6. Line 108, “with” should be replaced with “between”.
7. Line 147, the writing of “CRISPR/Cas9” should be unified in the whole manuscript.
8. Figure 4A, the PCR1 and PCR2 in the schematic diagram was incorrect.
9. Line 156-157, “Schematic diagram” and “model diagram” was repeat.
10. Figure 5C, the statistic analysis was missed in the figure.
11. Figure 6C, bottom panel should be blotted with Anti-HA rather than Anti-URM1, please correct.
12. Figure 6E legend, it will be better if the function of tBOOH was explain.
13. Line216-219, these two sentences are hard to understand.
14. Where is Figure 7G? I guess a wrong picture was uploaded, please correct.
Author Response
Dear editor:
Thank you for your comments, We have revised the manuscript based on the comments of the reviewers, and the details are as follows:
Point 1: Line 18, this sentence is misleading, please rephrase.
Response 1: Thank you for your suggestion. We have rephrase on lines 17-18.
Point 2: Line 21, please delete “only”.
Response 2: Modification was made following the reviewer's suggestion.
Point 3: Line 23, “reduced” was used incorrect, “attenuated” is better.
Response 3: We have changed "reduced" to "attenuated" on line 23.
Point 4: Line 71, How can the phylogenetic analysis come to a conclusion “indicating that TgUba1 can activate and modify TgUrm1”, it’s better to delete this sentence.
Response 4: Modification was made following the reviewer's suggestion.
Point 5: Line 85, “multi-antibody” is misleading, please rephrase.
Response 5: Thank you for your suggestion. We have changed "multi-antibody" to "polyclonal antibody" on line 89.
Point 6: Line 108, “with” should be replaced with “between”.
Response 6: We have changed "with" to "between" on line 112.
Point 7: Line 147, the writing of “CRISPR/Cas9” should be unified in the whole manuscript.
Response 7: The writing of “CRISPR/Cas9” has been unified in the whole manuscript following the reviewer's suggestion.
Point 8: Figure 4A, the PCR1 and PCR2 in the schematic diagram was incorrect.
Response 8: Thank you for your suggestion. We have corrected the schematic diagram of PCR1 and PCR2 in Figure 4A.
Point 9: Line 156-157, “Schematic diagram” and “model diagram” was repeat.
Response 9: Modification was made following the reviewer's suggestion.
Point 10: Figure 5C, the statistic analysis was missed in the figure.
Response 10: We've added data analytics following the reviewer's suggestion in Figure 5C.
Point 11: Figure 6C, bottom panel should be blotted with Anti-HA rather than Anti-URM1, please correct.
Response 11: Thank you for your suggestion. We have changed "Anti-URM1" to "Anti-HA" in Figure 6C.
Point 12: Figure 6E legend, it will be better if the function of tBOOH was explain.
Response 12: Thank you for your suggestion. We added the functional explanation of tBOOH in Figure 6E legend on line 216.
Point 13: Line216-219, these two sentences are hard to understand.
Response 13: We have revised and changed sentence on lines 218-221.
Point 14: Where is Figure 7G? I guess a wrong picture was uploaded, please correct.
Response 14: We have added Figure 7G following the reviewer's suggestion.
Reviewer 2 Report
The work provides new and important data on the molecular mechanisms of the medically important parasite Toxoplasma gondi. However, I have some comments on the form / structure of this article - some fragments from Results belong to Material and Methods or Discussion.
I recommend this paper for publication in the International Journal of Molecular Sciences after minor revisions.
1. Please provide the systematic positions of the taxa / species reported: Saccharomyces cerevisiae (Fungi), Kluyveromyces marxianus (…), Candida albicans (…).
2. Figure 1: please complete the generic names, because it is not known what species are meant, e.g. S. scrofa – Sus scrofa, etc, etc.
3. lines 107-108: „ To study the functional connection between TgUba1 and TgUrm1, we first used the yeast two-hybrid experiment to confirm the interaction with TgUba1 and TgUrm1” – this belongs to the Material and Methods.
4. Figure 1: Some things belong to the Material and Methods.
5. lines 126-127: “To further investigate the key domains mediating the interaction of TgUba1 with TgUrm1, yeast two-hybrid assays were employed to test the hypothesis.” - this belongs to the Material and Methods.
6. lines 131-132: “This interaction was also verified by Co-IP assay.” - this belongs to the Material and Methods.
7. Figure 13: Some things belong to the Material and Methods, e.g. “Immunoprecipitated samples were detected using western blots with anti-GFP and antiFlag mAb, respectively”.
8. lines 144-146: “To test the function of TgUba1 in the parasite, the CRISPR-Cas9 system was adopted to disrupt TgUba1 in the RHΔKu80 strain, thereby generating the knockout strain RHΔUba1 and the complementary strain CM-Uba1” - this belongs to the Material and Methods.
9. lines 153-154: “...as verified by diagnostic PCR (Fig. 4B), western blotting (Fig. 4C) and IFA”- this belongs to the Material and Methods.
10. line 165: ”The parasites of the ΔUba1 strain were used to infect DF-1 cells for plaque assay” - this belongs to the Material and Methods.
11. lines 168-169: “This suggests that ΔUba1 is defective in parasite proliferation, but not in invasion.” - this belongs to the Discussion.
12. lines 178-180: “These results demonstrate that the disruption of TgUba1 reduces Toxoplasma proliferation and virulence in mice.” - this belongs to the Discussion.
13. Figure 5: Some things belong to the Material and Methods, e.g “The results were analyzed by two-way ANOVA”, see paragraph 4.7. Statistical analysis !?
11. lines 198-199: “Previous studies in the laboratory found that oxidative stress induced covalent conjugation of URM1” – this belongs to the Discussion.
14. lines 216-222: “We next screened for other proteins in the specific T. gondii protein interactome that are expressed at endogenous levels, co-modify URM1 or exert other functions. To examine the extent to which TurboID biotinylates proteins in T. gondii, we generated CM-Uba1TurboID expressing a TurboID fusion protein based on ΔUba1 (Fig. 7A). PCR (Fig. 7B) and Western blotting (Fig. 7C) were selected for the screening and identification of monoclonal strains. As confirmed by IFA, 24 h biotin incubation of Uba1-TurboID tachyzoites resulted in robust biotinylation of the cellular proteome” - this belongs to the Material and Methods.
15. lines 224-225: “The differential protein bands were then subjected to protein identification by mass spectrometry’ - this belongs to the Material and Methods.
16. lines 228-230: “Taken together, our findings suggest that TgUba1 may be involved in ubiquitin-like modifications and anti-oxidative stress in T. gondii.” - this belongs to the Discussion.
17. Figure 7: Some things belong to the Material and Methods.
18. Figure 7: I can’t see “(G) Analysis of mass spectrometry data” ?
19. Line 310-311: “The T. gondii type I RH ΔKu80 strain was maintained in monkey kidney adherent ,epithelial (Vero)”- worth mentioning the scientific name of the monkey species - Chlorocebus sabaeus?
Author Response
Dear editor:
Thank you for your comments, We have revised the manuscript based on the comments of the reviewers, and the details are as follows:
Point 1: Please provide the systematic positions of the taxa / species reported: Saccharomyces cerevisiae (Fungi), Kluyveromyces marxianus (…), Candida albicans (…).
Response 1: Thank you for your suggestion. We have added the systematic positions of the taxa/species on lines 94-101.
Point 2: Figure 1: please complete the generic names, because it is not known what species are meant, e.g. S. scrofa – Sus scrofa, etc, etc.
Response 2: Thank you for your suggestion. We have completed the generic names on lines 94-101.
Point 3: lines 107-108: To study the functional connection between TgUba1 and TgUrm1, we first used the yeast two-hybrid experiment to confirm the interaction with TgUba1 and TgUrm1” – this belongs to the Material and Methods.
Response 3: We have revised and changed sentence on lines 111-113.
Point 4: Figure 1: Some things belong to the Material and Methods.
Response 4: Modification was made in Figure 1 following the reviewer's suggestion.
Point 5: lines 126-127: “To further investigate the key domains mediating the interaction of TgUba1 with TgUrm1, yeast two-hybrid assays were employed to test the hypothesis.” - this belongs to the Material and Methods.
Response 5: Modification was made on lines 131-133 following the reviewer's suggestion.
Point 6: lines 131-132: “This interaction was also verified by Co-IP assay.” - this belongs to the Material and Methods.
Response 6: Modification was made on line 136 following the reviewer's suggestion.
Point 7: Figure 13: Some things belong to the Material and Methods, e.g. “Immunoprecipitated samples were detected using western blots with anti-GFP and antiFlag mAb, respectively”.
Response 7: We have adjusted relevant statement to Materials and Methods section.
Point 8: lines 144-146: “To test the function of TgUba1 in the parasite, the CRISPR-Cas9 system was adopted to disrupt TgUba1 in the RHΔKu80 strain, thereby generating the knockout strain RHΔUba1 and the complementary strain CM-Uba1” - this belongs to the Material and Methods.
Response 8: We have changed the sentence to "To test the function of TgUba1 in the parasite, TgUba1 was disrupted in the RHΔKu80 strain, resulting in the knockout strain RHΔUba1 and the complementary strain CM-Uba1 " on lines 147-149.
Point 9: lines 153-154: “...as verified by diagnostic PCR (Fig. 4B), western blotting (Fig. 4C) and IFA”- this belongs to the Material and Methods.
Response 9: We have changed the sentence to "original genes existence was not detected by diagnostic PCR (Fig. 4B) and the proteins expression was not observed by Western blot (Fig. 4C) and IFA (Fig. 4D), confirming that we have obtained the knockout and complementary expression of tachyzoites" on lines 154-157.
Point 10: line 165: ”The parasites of the ΔUba1 strain were used to infect DF-1 cells for plaque assay” - this belongs to the Material and Methods.
Response 10: We have changed the sentence to "Obvious plaques difference were observed in parasites of ΔUba1 strain and CM-Uba1 strain" on lines 169-170.
Point 11: lines 168-169: “This suggests that ΔUba1 is defective in parasite proliferation, but not in invasion.” - this belongs to the Discussion.
Response 11: We have changed the sentence to "Briefly, plaque experiments demonstrated that ΔUba1 is defective in parasite proliferation, but not in invasion" on lines 172-173.
Point 12: lines 178-180: “These results demonstrate that the disruption of TgUba1 reduces Toxoplasma proliferation and virulence in mice.” - this belongs to the Discussion.
Response 12: We have changed the sentence to "In general, disruption of TgUba1 reduced T. gondii proliferation and pathogenicity in mice without affecting invasion and gliding motility" on lines 183-185.
Point 13: Figure 5: Some things belong to the Material and Methods, e.g “The results were analyzed by two-way ANOVA”, see paragraph 4.7. Statistical analysis !?
Response 13: We have revised and changed sentence on lines 371-374.
Point 14: lines 198-199: “Previous studies in the laboratory found that oxidative stress induced covalent conjugation of URM1” – this belongs to the Discussion.
Response 14: Thank you for your suggestion. Modification was made following the reviewer's suggestion.
Point 15: lines 216-222: “We next screened for other proteins in the specific T. gondii protein interactome that are expressed at endogenous levels, co-modify URM1 or exert other functions. To examine the extent to which TurboID biotinylates proteins in T. gondii, we generated CM-Uba1TurboID expressing a TurboID fusion protein based on ΔUba1 (Fig. 7A). PCR (Fig. 7B) and Western blotting (Fig. 7C) were selected for the screening and identification of monoclonal strains. As confirmed by IFA, 24 h biotin incubation of Uba1-TurboID tachyzoites resulted in robust biotinylation of the cellular proteome” - this belongs to the Material and Methods.
Response 15: We have changed the sentence to "Except the two classical domains, TgUba1 has a special domain-ANT1. To investigate other functions exerted of TgUba1 in T. gondii, CM-Uba1-TurboID strain expressing a TurboID fusion protein was generated (Fig. 7A). PCR showed a correct integration of TurboID at the C-terminus of TgUba1 (Fig. 7B). A 173kDa band was observed in Western blot indicating the fusion expression of TgUba1 and TurboID (Fig. 7C). In Figure 7D, the interacting proteins of TurboID-tagged TgUba1 was detected to accumulate in the cytoplasm" on lines 219-225.
Point 16: lines 224-225: “The differential protein bands were then subjected to protein identification by mass spectrometry’ - this belongs to the Material and Methods.
Response 16: Modification was made on lines 227-228 following the reviewer's suggestion.
Point 17: lines 228-230: “Taken together, our findings suggest that TgUba1 may be involved in ubiquitin-like modifications and anti-oxidative stress in T. gondii.” - this belongs to the Discussion.
Response 17: Modification was made following the reviewer's suggestion.
Point 18: Figure 7: Some things belong to the Material and Methods.
Response 18: Modification was made following the reviewer's suggestion in Figure 7.
Point 19: Figure 7: I can’t see “(G) Analysis of mass spectrometry data” ?
Response 19: We have re-uploaded Figure 7 to supplement the missing.
Point 20: Line 310-311: “The T. gondii type I RH ΔKu80 strain was maintained in monkey kidney adherent ,epithelial (Vero)”- worth mentioning the scientific name of the monkey species - Chlorocebus sabaeus?
Response 20: We have revised and changed sentence on line 314.
Reviewer 3 Report
In the current study Xiao and coworkers have identified TgUba1 as a bona fide interacting partner of Urm1 and further show that this specific interaction is required for processing of Urm1 in vivo. The authors show that TgUba1 is required for the parasite proliferation in both cell and mouse model. Furthermore, they show that cells lacking TgUba1 is defective in eliciting Urm1 mediated oxidative stress response. Altogether the authors propose a model in which TgUba1 acts like E1-ligase and is required for Urm1 activation – an essential step in the Urm modification of the proteins involved in the growth and proliferation of the parasite during infection cycle. The model is substantially supported by the data, and I would like to congratulate the authors.
There are some minor points.
1. Please mention the number of mice used in the experiment and how many times the experiment was repeated.
2. Some graph legends in the figure are hard to read when printed. You could move some control experiments to the supporting information or increase the size of the figure, as directed by the journal guidelines.
3. The manuscript is written in an esoteric fashion; It will be helpful for the readers if you could elaborate certain sections, both in introduction and results.
4. In future for Mass-spec experiment you could AD-mutant as a control to get confident and specific hits.
Author Response
Dear editor:
Thank you for your comments, We have revised the manuscript based on the comments of the reviewers, and the details are as follows:
Point 1: Please mention the number of mice used in the experiment and how many times the experiment was repeated.
Response 1: We have added a description of the mice pathogenicity test on lines 331-335.
Point 2: Some graph legends in the figure are hard to read when printed. You could move some control experiments to the supporting information or increase the size of the figure, as directed by the journal guidelines.
Response 2: We have resized the images and modified some legends to make them more understandable, such as figure 1 and figure 5.
Point 3: The manuscript is written in an esoteric fashion; It will be helpful for the readers if you could elaborate certain sections, both in introduction and results.
Response 3: Thank you for your suggestion. We have added some necessary information in the introduction on lines 60-63.
Point 4: In future for Mass-spec experiment you could AD-mutant as a control to get confident and specific hits.
Response 4: The reviewer provided rigorous ideas for our future experiments. Fortunately, we have screened some interacting proteins with extensive experimental verification. Certainly, we will follow the reviewer's suggestion to set more controls in the future to expect more accurate experimental results.